# Separation and Concentration in Deep Networks

**John Zarka, Florentin Guth**
Département d'informatique de l'ENS, ENS, CNRS, PSL University, Paris, France
`{john.zarka,florentin.guth}@ens.fr`

**Stéphane Mallat**
Collège de France, Paris, France
Flatiron Institute, New York, USA

## Abstract

Numerical experiments demonstrate that deep neural network classifiers progressively separate class distributions around their mean, achieving linear separability on the training set, and increasing the Fisher discriminant ratio. We explain this mechanism with two types of operators. We prove that a rectifier without biases applied to sign-invariant tight frames can separate class means and increase Fisher ratios. On the opposite, a soft-thresholding on tight frames can reduce within-class variabilities while preserving class means. Variance reduction bounds are proved for Gaussian mixture models. For image classification, we show that separation of class means can be achieved with rectified wavelet tight frames that are not learned. It defines a scattering transform. Learning $1 \times 1$ convolutional tight frames along scattering channels and applying a soft-thresholding reduces within-class variabilities. The resulting scattering network reaches the classification accuracy of ResNet-18 on CIFAR-10 and ImageNet, with fewer layers and no learned biases.

## 1 Introduction

Several numerical works (Oyallon, 2017; Papyan, 2020; Papyan et al., 2020) have shown that deep neural networks classifiers (LeCun et al., 2015) progressively concentrate each class around separated means, until the last layer, where within-classes variability may nearly "collapse" (Papyan et al., 2020). The linear separability of a class mixture is characterized by the Fisher discriminant ratio (Fisher, 1936; Rao, 1948). The Fisher discriminant ratio measures the separation of class means relatively to the variability within each class, as measured by their covariances. The neural collapse appears through a considerable increase of the Fisher discriminant ratio during training (Papyan et al., 2020). No mathematical mechanism has yet been provided to explain this separation and concentration of probability measures.

Linear separability and Fisher ratios can be increased by separating class means without increasing the variability of each class, or by concentrating each class around its mean while preserving the mean separation. This paper shows that these separation or concentration properties can be achieved with one-layer network operators using different pointwise non-linearities. We cascade these operators to define structured deep neural networks with high classification accuracies, and which can be analyzed mathematically.

Section 2 studies two-layer networks computed with a linear classifier applied to $\rho F$, where $F$ is linear and $\rho$ is a pointwise non-linearity. First, we show that $\rho F$ can separate class means with a ReLU $\rho_r(u) = \max(u, 0)$ and a sign-invariant $F$. We prove that $\rho_r F$ then increases the Fisher ratio. As in Parseval networks (Cisse et al., 2017), $F$ is normalized by imposing that it is a tight frame which satisfies $F^T F = \text{Id}$. Second, to concentrate the variability of each class around its mean, we use a shrinking non-linearity implemented by a soft-thresholding $\rho_t$. For Gaussian mixture models, we prove that $\rho_t F$ concentrates within-class variabilities while nearly preserving class means, under appropriate sparsity hypotheses. A linear classifier applied to these $\rho F$ defines two-layer

neural networks with no learned bias parameters in the hidden layer, whose properties are studied mathematically and numerically.

Cascading several convolutional tight frames with ReLUs or soft-thresholdings defines a deep neural network which progressively separates class means and concentrates their variability. One may wonder if we can avoid learning these frames by using prior information on the geometry of images. Section 3 shows that the class mean separation can be computed with wavelet tight frames, which are not learned. They separate scales, directions and phases, which are known groups of transformations. A cascade of wavelet filters and rectifiers defines a scattering transform (Mallat, 2012), which has previously been applied to image classification (Bruna & Mallat, 2013; Oyallon & Mallat, 2015). However, such networks do not reach state-of-the-art classification results. We show that important improvements are obtained by learning $1 \times 1$ convolutional projectors and tight frames, which concentrate within-class variabilities with soft-thresholdings. It defines a bias-free deep scattering network whose classification accuracy reaches ResNet-18 (He et al., 2016) on CIFAR-10 and ImageNet. Code to reproduce all experiments of the paper is available at `https://github.com/j-zarka/separation_concentration_deepnets`.

The main contributions of this paper are:

- A double mathematical mechanism to separate and concentrate distinct probability measures, with a rectifier and a soft-thresholding applied to tight frames. The increase of Fisher ratio is proved for tight-frame separation with a rectifier. Bounds on within-class covariance reduction are proved for a soft-thresholding on Gaussian mixture models.

- The introduction of a bias-free scattering network which reaches ResNet-18 accuracy on CIFAR-10 and ImageNet. Learning is reduced to $1 \times 1$ convolutional tight frames which concentrate variabilities along scattering channels.

## 2    CLASSIFICATION BY SEPARATION AND CONCENTRATION

The last hidden layer of a neural network defines a representation $\Phi(x)$, to which is applied a linear classifier. This section studies the separation of class means and class variability concentration for $\Phi = \rho F$ in a two-layer network.

### 2.1    TIGHT FRAME RECTIFICATION AND THRESHOLDING

We begin by briefly reviewing the properties of linear classifiers and Fisher discriminant ratios. We then analyze the separation and concentration of $\Phi = \rho F$, when $\rho$ is a rectifier or a soft-thresholding and $F$ is a tight frame.

**Linear classification and Fisher ratio**    We consider a random data vector $x \in \mathbb{R}^d$ whose class labels are $y(x) \in \{1, ..., C\}$. Let $x_c$ be a random vector representing the class $c$, whose probability distribution is the distribution of $x$ conditioned by $y(x) = c$. We suppose that all classes are equiprobable for simplicity. $\mathrm{Ave}_c$ denotes $C^{-1} \sum_{c=1}^{C}$.

We compute a representation of $x$ with an operator $\Phi$ which is standardized, so that $\mathbb{E}(\Phi(x)) = 0$ and each coefficient of $\Phi(x)$ has a unit variance. The class means $\mu_c = \mathbb{E}(\Phi(x_c))$ thus satisfy $\sum_c \mu_c = 0$. A linear classifier $(W, b)$ on $\Phi(x)$ returns the index of the maximum coordinate of $W\Phi(x) + b \in \mathbb{R}^C$. An optimal linear classifier $(W, b)$ minimizes the probability of a classification error. Optimal linear classifiers are estimated by minimizing a regularized loss function on the training data. Neural networks often use logistic linear classifiers, which minimize a cross-entropy loss. The standardization of the last layer $\Phi(x)$ is implemented with a batch normalization (Ioffe & Szegedy, 2015).

A linear classifier can have a small error if the typical sets of each $\Phi(x_c)$ have little overlap, and in particular if the class means $\mu_c = \mathbb{E}(\Phi(x_c))$ are sufficiently separated relatively to the variability of each class. Under the Gaussian hypothesis, the variability of each class is measured by the covariance $\Sigma_c$ of $\Phi(x_c)$. Let $\Sigma_W = \mathrm{Ave}_c \, \Sigma_c$ be the average within-class covariance and $\Sigma_B = \mathrm{Ave}_c \, \mu_c \, \mu_c^T$ be the between-class covariance of the means. The within-class covariance can be whitened and normalized to Id by transforming $\Phi(x)$ with the square root $\Sigma_W^{-\frac{1}{2}}$ of $\Sigma_W^{-1}$. All classes

$c, c'$ are highly separated if $\|\Sigma_W^{-\frac{1}{2}}\mu_c - \Sigma_W^{-\frac{1}{2}}\mu_{c'}\| \gg 1$. This separation is captured by the Fisher discriminant ratio $\Sigma_W^{-1}\Sigma_B$. We shall measure its trace:

$$C^{-1}\,\mathrm{Tr}(\Sigma_W^{-1}\Sigma_B) = \underset{c}{\mathrm{Ave}}\,\|\Sigma_W^{-\frac{1}{2}}\mu_c\|^2. \tag{1}$$

Fisher ratios have been used to train deep neural networks as a replacement for the cross-entropy loss (Dorfer et al., 2015; Stuhlsatz et al., 2012; Sun et al., 2019; Wu et al., 2017; Sultana et al., 2018; Li et al., 2016). In this paper, we use their analytic expression to analyze the improvement of linear classifiers.

Linear classification obviously cannot be improved with a linear representation $\Phi$. The following proposition gives a simple condition to improve (or maintain) the error of linear classifiers with a non-linear representation.

**Proposition 2.1.** *If $\Phi$ has a linear inverse, then it decreases (or maintains) the error of the optimal linear classifier, and it increases (or maintains) the Fisher ratio (1).*

To prove this result, observe that if $\Phi$ has a linear inverse $\Phi^{-1}$ then $Wx = W'\Phi(x)$ with $W' = W\Phi^{-1}$. The minimum classification error by optimizing $W$ is thus above the error obtained by optimizing $W'$. Appendix A proves that the Fisher ratio (1) is also increased or preserved.

There are qualitatively two types of non-linear operators that increase the Fisher ratio $\Sigma_W^{-1}\Sigma_B$. Separation operators typically increase the distance between the class means without increasing the variance $\Sigma_W$ within each class. We first study such operators having a linear inverse, which guarantees through Proposition 2.1 that they increase the Fisher ratio. We then study concentration operators which reduce the variability $\Sigma_W$ with non-linear shrinking operators, which are not invertible. It will thus require a finer analysis of their properties.

**Separation by tight frame rectification**  Let $\Phi = \rho F$ be an operator which computes the first layer of a neural network, where $\rho$ is a pointwise non-linearity and $F$ is linear. We first study separation operators computed with a ReLU $\rho_r(u) = \max(u, 0)$ applied to an invertible sign-invariant matrix. Such a matrix has rows that can be regrouped in pairs of opposite signs. It can thus be written $F = [-\tilde{F}^T, \tilde{F}^T]^T$ where $\tilde{F}$ is invertible. The operator $\rho F$ separates coefficients according to their sign. Since $\rho_r(u) - \rho_r(-u) = u$, it results that $\Phi = \rho_r F$ is linearly invertible. According to Proposition 2.1, it increases (or maintains) the Fisher ratio, and we want to choose $F$ to maximize this increase.

Observe that $\rho_r(\alpha u) = \alpha \rho_r(u)$ if $\alpha \geq 0$. We can thus normalize the rows $f_m$ of $F$ without affecting linear classification performance. To ensure that $F \in \mathbb{R}^{p \times d}$ is invertible with a stable inverse, we impose that it is a normalized tight frame of $\mathbb{R}^d$ satisfying

$$F^T F = \mathrm{Id} \;\; \text{and} \;\; \|f_m\|^2 = d/p \;\text{for}\; 1 \leq m \leq p.$$

The tight frame can be interpreted as a rotation operator in a higher dimensional space, which aligns the axes and the directions along which $\rho_r$ performs the sign separation. This rotation must be adapted in order to optimize the separation of class means. The fact that $F$ is a tight frame can be interpreted as a normalization which simplifies the mathematical analysis.

Suppose that all classes $x_c$ of $x$ have a Gaussian distribution with a zero mean $\mu_c = 0$, but different covariances $\Sigma_c$. These classes are not linearly separable because they have the same mean, and the Fisher ratio is $0$. Applying $\rho_r F$ can separate these classes and improve the Fisher ratio. Indeed, if $z$ is a zero-mean Gaussian random variable, then $\mathbb{E}(\max(z, 0)) = (2\pi)^{-1/2}\mathbb{E}(z^2)^{1/2}$ so we verify that for $F = [-\tilde{F}^T, \tilde{F}^T]^T$,

$$\mathbb{E}(\rho_r F x_c) = (2\pi)^{-1/2}\Big(\mathrm{diag}(\tilde{F}\Sigma_c\tilde{F}^T)^{1/2}, \mathrm{diag}(\tilde{F}\Sigma_c\tilde{F}^T)^{1/2}\Big).$$

The Fisher ratio can then be optimized by maximizing the covariance $\Sigma_B$ between the mean vector components $\mathrm{diag}(\tilde{F}\Sigma_c F^T)^{1/2}$ for all classes $c$. If we know a priori that that $x_c$ and $-x_c$ have the same probability distribution, as in the Gaussian example, then we can replace $\rho_r$ by the absolute value $\rho_a(u) = |u| = \rho_r(u) + \rho_r(-u)$, and $\rho_r F$ by $\rho_a \tilde{F}$, which reduces by 2 the frame size.

**Concentration by tight frame soft-thresholding**   If the class means of $x$ are already separated, then we can increase the Fisher ratio with a non-linear $\Phi$ that concentrates each class around its mean. The operator $\Phi$ must reduce the within-class variance while preserving the class separation. This can be interpreted as a non-linear noise removal if we consider the within-class variability as an additive noise relatively to the class mean. It can be done with soft-thresholding estimators introduced in Donoho & Johnstone (1994). A soft-thresholding $\rho_t(u) = \text{sign}(u) \max(|u| - \lambda, 0)$ shrinks the amplitude of $u$ by $\lambda$ in order to reduce its variance, while introducing a bias that depends on $\lambda$. Donoho & Johnstone (1994) proved that soft-thresholding estimators are highly effective to estimate signals that have a sparse representation in a tight frame $F$.

To evaluate more easily the effect of a tight frame soft-thresholding on the class means, we apply the linear reconstruction $F^T$ on $\rho_t F x$, which thus defines a representation $\Phi(x) = F^T \rho_t F x$. For a strictly positive threshold, this operator is not invertible, so we cannot apply Proposition 2.1 to prove that the Fisher ratio increases. We study directly the impact of $\Phi$ on the mean and covariance of each class. Let $x_c$ be the vector representing the class $c$. The mean $\mu_c = \mathbb{E}(x_c)$ is transformed into $\bar{\mu}_c = \mathbb{E}(\Phi(x_c))$ and the covariance $\Sigma_c$ of $x_c$ into the covariance $\overline{\Sigma}_c$ of $\Phi(x_c)$. The average covariances are $\Sigma_W = \text{Ave}_c \, \Sigma_c$ and $\overline{\Sigma}_W = \text{Ave}_c \, \overline{\Sigma}_c$.

Suppose that each $x_c$ is a Gaussian mixture, with a potentially large number of Gaussian components centered at $\mu_{c,k}$ with a fixed covariance $\sigma^2 \text{Id}$:

$$p_c = \sum_k \pi_{c,k} \, \mathcal{N}(\mu_{c,k}, \sigma^2 \text{Id}). \tag{2}$$

This model is quite general, since it amounts to covering the typical set of realizations of $x_c$ with a union of balls of radius $\sigma$, centered in the $(\mu_{c,k})_k$. The following theorem relates the reduction of within-class covariance to the sparsity of $F \mu_{c,k}$. It relies on the soft-thresholding estimation results of Donoho & Johnstone (1994).

For simplicity, we suppose that the tight frame is an orthogonal basis, but the result can be extended to general normalized tight frames. The sparsity is expressed through the decay of sorted basis coefficients. For a vector $z \in \mathbb{R}^d$, we denote $z^{(r)}$ a coefficient of rank $r$: $|z^{(r)}| \geq |z^{(r+1)}|$ for $1 \leq r \leq d$. The theorem imposes a condition on the amplitude decay of the $(F\mu_{c,k})^{(r)}$ when $r$ increases, which is a sparsity measure. We write $a(r) \sim b(r)$ if $C_1 \, a(r) \leq b(r) \leq C_2 \, a(r)$ where $C_1$ and $C_2$ do not depend upon $d$ nor $\sigma$. The theorem derives upper bounds on the reduction of within-class covariances and on the displacements of class means. The constants do not depend upon $d$ when it increases to $\infty$ nor on $\sigma$ when it decreases to 0.

**Theorem 2.2.** *Under the mixture model hypothesis (2), we have:*

$$\text{Tr}(\Sigma_W) = \text{Tr}(\Sigma_M) + \sigma^2 \, d, \;\; \text{with} \;\; \text{Tr}(\Sigma_M) = C^{-1} \sum_{c,k} \pi_{c,k} \, \|\mu_c - \mu_{c,k}\|^2. \tag{3}$$

*If there exists $s > 1/2$ such that $|(F\mu_{c,k})^{(r)}| \sim r^{-s}$ then a tight frame soft-thresholding with threshold $\lambda = \sigma \sqrt{2 \log d}$ satisfies:*

$$\text{Tr}(\overline{\Sigma}_W) = 2 \, \text{Tr}(\Sigma_M) + O(\sigma^{2-1/s} \log d), \tag{4}$$

*and all class means satisfy:*

$$\|\mu_c - \bar{\mu}_c\|^2 = O(\sigma^{2-1/s} \log d). \tag{5}$$

Under appropriate sparsity hypotheses, the theorem proves that applying $\Phi = F^T \rho_t F$ reduces considerably the trace of the within-class covariance. The Gaussian variance $\sigma^2 d$ is dominant in (3) and is reduced to $O(\sigma^{2-1/s} \log d)$ in (4). The upper bound (5) also proves that $F^T \rho_t F$ creates a relatively small displacement of class means, which is proportional to $\log d$. This is important to ensure that all class means remain well separated. These bounds qualitatively explains the increase of Fisher ratios, but they are not sufficient to prove a precise bound on these ratios.

In numerical experiments, the threshold value of the theorem is automatically adjusted as follows. Non-asymptotic optimal threshold values have been tabulated as a function of $d$ by Donoho & Johnstone (1994). For the range of $d$ used in our applications, a nearly optimal threshold is $\lambda = 1.5 \, \sigma$. We rescale the frame variance $\sigma^2$ by standardizing the input $x$ so that it has a zero mean and each coefficient has a unit variance. In high dimension $d$, the within-class variance typically dominates

the variance between class means. Under the unit variance assumption we have $\mathrm{Tr}(\Sigma_W) \approx d$. If $F \in \mathbb{R}^{p \times d}$ is a normalized tight frame then we also verify as in (3) that $\mathrm{Tr}(\Sigma_W) \approx \sigma^2 p$ so $\sigma^2 \approx d/p$. It results that we choose $\lambda = 1.5 \sqrt{d/p}$.

A soft-thresholding can also be computed from a ReLU with threshold $\rho_{rt}(u) = \max(u - \lambda, 0)$ because $\rho_t(u) = \rho_{rt}(u) - \rho_{rt}(-u)$. It results that $[F^T, -F^T] \rho_{rt} [F^T, -F^T]^T = F^T \rho_t F$. However, a thresholded rectifier has more flexibility than a soft-thresholding, because it may recombine differently $\rho_{rt}F$ and $\rho_{rt}(-F)$ to also separate class means, as explained previously. The choice of threshold then becomes a trade-off between separation of class means and concentration of class variability. In numerical experiments, we choose a lower $\lambda = \sqrt{d/p}$ for a ReLU with a threshold.

## 2.2 Two-Layer Networks without Bias

We study two-layer bias-free networks that implement a linear classification on $\rho F$, where $F$ is a normalized tight frame and $\rho$ may be a rectifier, an absolute value or a soft-thresholding, with no learned bias parameter. Bias-free networks have been introduced for denoising in Mohan et al. (2019), as opposed to classification or regression. We show that such bias-free networks have a limited expressivity and do *not* satisfy universal approximation theorems (Pinkus, 1999; Bach, 2017). However, numerical results indicate that their separation and contractions capabilities are sufficient to reach similar classification results as two-layer networks with biases on standard image datasets.

Applying a linear classifier on $\Phi(x)$ computes:

$$W\Phi(x) + b = W\rho F x + b.$$

This two-layer neural network has no learned bias parameters in the hidden layer, and we impose that $F^T F = \mathrm{Id}$ with frame rows $(f_m)_m$ having constant norms. As a result, the following theorem proves that it does not satisfy the universal approximation theorem. We define a binary classification problem for which the probability of error remains above $1/4$ for any number $p$ of neurons in the hidden layer. The proof is provided in Appendix C for a ReLU $\rho_{rt}$ with any threshold. The theorem remains valid with an absolute value $\rho_a$ or a soft-thresholding $\rho_t$, because they are linear combinations of $\rho_{rt}$.

**Theorem 2.3.** *Let $\lambda \geq 0$ be a fixed threshold and $\rho_{rt}(u) = \max(u - \lambda, 0)$. Let $\mathcal{F}$ be the set of matrices $F \in \mathbb{R}^{p \times d}$ with bounded rows $\|f_m\| \leq 1$. There exists a random vector $x \in \mathbb{R}^d$ which admits a probability density supported on the unit ball, and a $C^\infty$ function $h \colon \mathbb{R}^d \to \mathbb{R}$ such that, for all $p \geq d$:*

$$\inf_{W \in \mathbb{R}^{1 \times p}, F \in \mathcal{F}, b \in \mathbb{R}} \mathbb{P}[\mathrm{sgn}(W\rho_{rt}Fx + b) \neq \mathrm{sgn}(h(x))] \geq \frac{1}{4}.$$

**Optimization** The parameters $W$, $F$ and $b$ are optimized with a stochastic gradient descent that minimizes a logistic cross-entropy loss on the output. To impose $F^T F = \mathrm{Id}$, following the optimization of Parseval networks (Cisse et al., 2017), after each gradient update of all network parameters, we insert a second gradient step to minimize $\alpha/2 \|F^T F - \mathrm{Id}\|^2$. This gradient update is:

$$F \leftarrow (1 + \alpha)F - \alpha F F^T F. \tag{6}$$

We also make sure after every Parseval step that each tight frame row $f_m$ keeps a constant norm $\|f_m\| = \sqrt{d/p}$ by applying a spherical projection: $f_m \leftarrow \sqrt{d/p} \, f_m / \|f_m\|$. These steps are performed across all experiments described in the paper, which ensures that all singular values of every learned tight frame are comprised between 0.99 and 1.01.

To reduce the number of parameters of the classification matrix $W \in \mathbb{R}^{C \times p}$, we factorize $W = W' F^T$ with $W' \in \mathbb{R}^{C \times d}$. It amounts to reprojecting $\rho F$ in $\mathbb{R}^d$ with the semi-orthogonal frame synthesis $F^T$, and thus defines:

$$\Phi(x) = F^T \rho F x.$$

A batch normalization is introduced after $\Phi$ to stabilize the learning of $W'$.

**Image classification by separation and concentration** Image classification is first evaluated on the MNIST (LeCun et al., 2010) and CIFAR-10 (Krizhevsky, 2009) image datasets. Table 1 gives

Table 1: For MNIST and CIFAR-10, the first row gives the logistic classification error and the second row the Fisher ratio (1), for different signal representations $\Phi(x)$. Results are evaluated with an absolute value $\rho_a$, a soft-thresholding $\rho_t$, and a ReLU with threshold $\rho_{rt}$.

|  | $\Phi(x)$ | $x$ | $F^T \rho F x$ | | | $S_T(x)$ |
|---|---|---|---|---|---|---|
|  |  |  | $\rho = \rho_a$ | $\rho = \rho_t$ | $\rho = \rho_{rt}$ |  |
| **MNIST** | Error (%) | 7.4 | 1.3 | 1.4 | 1.3 | 0.8 |
|  | Fisher | 19 | 68 | 69 | 67 | 130 |
| **CIFAR** | Error (%) | 60.5 | 28.1 | 34.8 | 26.5 | 27.7 |
|  | Fisher | 6.7 | 15 | 13 | 16 | 12 |

the results of logistic classifiers applied to the input signal $x$ and to $\Phi(x) = F^T \rho F x$ for 3 different non-linearities $\rho$: absolute value $\rho_a$, soft-thresholding $\rho_t$, and ReLU with threshold $\rho_{rt}$. The tight frame $F$ is a convolution on patches of size $k \times k$ with a stride of $k/2$, with $k = 14$ for MNIST and $k = 8$ for CIFAR. The tight frame $F$ maps each patch to a vector of larger dimension, specified in Appendix D. Figure 1 in Appendix D shows examples of learned tight frame filters.

On each dataset, applying $F^T \rho F$ on $x$ greatly reduces linear classification error, which also appears with an increase of the Fisher ratio. For MNIST, all non-linearities produce nearly the same classification accuracy, but on CIFAR, the soft-thresholding has a higher error. Indeed, the class means of MNIST are distinct averaged digits, which are well separated, because all digits are centered in the image. Concentrating variability with a soft-thresholding is then sufficient. On the opposite, the classes of CIFAR images define nearly stationary random vectors because of arbitrary translations. As a consequence, the class means $\mu_c$ are nearly constant images, which are only discriminated by their average color. Separating these class means is then important for improving classification. As explained in Section 2.1, this is done by a ReLU $\rho_r$, or in this case an absolute value $\rho_a$, which reduces the error. The ReLU with threshold $\rho_{rt}$ can interpolate between mean separation and variability concentration, and thus performs usually at least as well as the other non-linearities.

The error of the bias-free networks with a ReLU and an absolute value are similar to the errors obtained by training two-layer networks of similar sizes but with bias parameters: 1.6% error on MNIST (Simard et al., 2003), and 25% on CIFAR-10 (Krizhevsky, 2010). It indicates that the elimination of bias parameters does not affect performances, despite the existence of the counter-examples from Theorem 2.3 that cannot be well approximated by such architectures. This means that image classification problems have more structure that are not captured by these counter-examples, and that completeness in linear high-dimensional functional spaces may not be key mathematical properties to explain the preformances of neural networks. Figure 1 in Appendix D shows that the learned convolutional tight frames include oriented oscillatory filters, which is also often the case of the first layer of deeper networks (Krizhevsky et al., 2012). They resemble wavelet frames, which are studied in the next section.

## 3 DEEP LEARNING BY SCATTERING AND CONCENTRATING

To improve classification accuracy, we cascade mean separation and variability concentration operators, implemented by ReLUs and soft-thresholdings on tight frames. This defines deep convolutional networks. However, we show that some tight frames do not need to be learned. Section 3.1 reviews scattering trees, which perform mean separation by cascading ReLUs on wavelet tight frames. Section 3.2 shows that we reach high classification accuracies by learning projectors and tight frame soft-thresholdings, which concentrate within-class variabilities along scattering channels.

### 3.1 SCATTERING CASCADE OF WAVELET FRAME SEPARATIONS

Scattering transforms have been introduced to classify images by cascading predefined wavelet filters with a modulus or a rectifier non-linearity (Bruna & Mallat, 2013). We write it as a product of wavelet tight frame rectifications, which progressively separate class means.

**Wavelet frame** A wavelet frame separates image variations at different scales, directions and phases, with a cascade of filterings and subsamplings. We use steerable wavelets (Simoncelli & Freeman, 1995) computed with Morlet filters (Bruna & Mallat, 2013). There is one low-pass filter $g_0$, and $L$ complex band-pass filters $g_\ell$ having an angular direction $\theta = \ell\pi/L$ for $0 < \ell \leq L$. These filters can be adjusted (Selesnick et al., 2005) so that the filtering and subsampling:

$$\tilde{F}_w x(n, \ell) = x \star g_\ell(2n)$$

defines a complex tight frame $\tilde{F}_w$. Fast multiscale wavelet transforms are computed by cascading the filter bank $\tilde{F}_w$ on the output of the low-pass filter $g_0$ (Mallat, 2008).

Each complex filter $g_\ell$ is analytic, and thus has a real part and imaginary part whose phases are shifted by $\alpha = \pi/2$. This property is important to preserve equivariance to translation despite the subsampling with a stride of 2 (Selesnick et al., 2005). To define a sign-invariant frame as in Section 2.1, we must incorporate filters of opposite signs, which amounts to shifting their phase by $\pi$. We thus associate to $\tilde{F}_w$ a real sign-invariant tight frame $F_w$ by considering separately the four phases $\alpha = 0, \pi/2, \pi, 3\pi/2$. It is defined by

$$F_w x(n, \ell, \alpha) = x \star g_{\ell,\alpha}(2n),$$

with $g_{\ell,0} = 2^{-1/2}\mathrm{Real}(g_\ell)$, $g_{\ell,\pi/2} = 2^{-1/2}\mathrm{Imag}(g_\ell)$ and $g_{\ell,\alpha+\pi} = -g_\ell$. We apply a rectifier $\rho_r$ to the output of all real band-pass filters $g_{\ell,\alpha}$ but not to the low-pass filter:

$$\rho_r F_w = \Big(x \star g_0(2n)\,,\; \rho_r(x \star g_{\ell,\alpha}(2n))\Big)_{n,\alpha,\ell}.$$

The use of wavelet phase parameters with rectifiers is studied in Mallat et al. (2019). The operator $\rho_r F_w$ is linearly invertible because $F_w$ is a tight frame and the ReLU is applied to band-pass filters, which come in pairs of opposite sign. Since there are 4 phases and a subsampling with a stride of 2, $F_w x$ is $(L + 1/4)$ times larger than $x$.

**Scattering tree** A full scattering tree $S_T$ of depth $J$ is computed by iterating $J$ times over $\rho_r F_w$. Since each $\rho_r F_w$ has a linear inverse, Proposition 2.1 proves that this separation can only increase the Fisher ratio. However it also increases the signal size by $(L + 1/4)^J$, which is typically much too large. This is avoided with orthogonal projectors, which perform a dimension reduction after applying each $\rho_r F_w$.

A pruned scattering tree $S_T$ of depth $J$ and order $o$ is defined in Bruna & Mallat (2013) as a convolutional tree which cascades $J$ rectified wavelet filter banks, and at each depth prunes the branches with $P_j$ to prevent an exponential growth:

$$S_T = \prod_{j=1}^{J} P_j\, \rho_r\, F_w. \tag{7}$$

After the ReLU, the pruning operator $P_j$ eliminates the branches of the scattering which cascade more than $o$ band-pass filters and rectifiers, where $o$ is the scattering order (Bruna & Mallat, 2013). After $J$ cascades, the remaining channels have thus been filtered by at least $J - o$ successive low-pass filters $g_0$. We shall use a scattering transform of order $o = 2$. The operator $P_j$ also averages the rectified output of the filters $g_{\ell,\alpha}$ along the phase $\alpha$, for $\ell$ fixed. This averaging eliminates the phase. It approximatively computes a complex modulus and produces a localized translation invariance. The resulting pruning and phase average operator $P_j$ is a $1 \times 1$ convolutional operator, which reduces the dimension of scattering channels with an orthogonal projection. If $x$ has $d$ pixels, then $S_T(x)[n, k]$ is an array of images having $2^{-2J}d$ pixels at each channel $k$, because of the $J$ subsamplings with a stride of 2. The total number of channels $K$ is $1 + JL + J(J - 1)L^2/2$. Numerical experiments are performed with wavelet filters which approximate Gabor wavelets (Bruna & Mallat, 2013), with $L = 8$ directions. The number of scales $J$ depends upon the image size. It is $J = 3$ for MNIST and CIFAR, and $J = 4$ for ImageNet, resulting in respectively $K = 217, 651$ and $1251$ channels.

Each $\rho_r F_w$ can only improve the Fisher ratio and the linear classification accuracy, but it is not guaranteed that this remains valid after applying $P_j$. Table 1 gives the classification error of a logistic classifier applied on $S_T(x)$, after a $1 \times 1$ orthogonal projection to reduce the number of channels, and a spatial normalization. This error is almost twice smaller than a two-layer neural network on MNIST, given in Table 1, but it does not improve the error on CIFAR. On CIFAR, the error obtained by a ResNet-20 is 3 times lower than the one of a classifier on $S_T(x)$. The main issue is now to understand where this inefficiency comes from.

Table 2: Linear classification error and Fisher ratios (1) of several scattering representations, on CIFAR-10 and ImageNet. For $S_C$, results are evaluated with a soft-thresholding $\rho_t$ and a thresholded rectifier $\rho_{rt}$. The last column gives the error of ResNet-20 for CIFAR-10 (He et al., 2016) and ResNet-18 for ImageNet, taken from `https://pytorch.org/docs/stable/torchvision/models.html`.

|  | $\Phi$ |  | $S_T$ | $S_P$ | $S_C$ ($\rho_t$) | $S_C$ ($\rho_{rt}$) | ResNet |
|---|---|---|---|---|---|---|---|
| **CIFAR** | Error (%) |  | 27.7 | 12.8 | 8.0 | 7.6 | 8.8 |
|  | Fisher |  | 12 | 20 | 43 | 41 | - |
| **ImageNet** | Error (%) | Top-5 | 54.1 | 20.5 | 11.6 | 10.7 | 10.9 |
|  |  | Top-1 | 73.0 | 42.3 | 31.4 | 29.7 | 30.2 |
|  | Fisher |  | 2.0 | 18 | 51 | 44 | - |

## 3.2 Separation and Concentration in Learned Scattering Networks

A scattering tree iteratively separates class means with wavelet filters. Its dimension is reduced by predefined projection operators, which may decrease the Fisher ratio and linear separability. To avoid this source of inefficiency, we define a scattering network which learns these projections. The second step introduces tight frame thresholdings along scattering channels, to concentrate within-class variabilities. Image classification results are evaluated on the CIFAR-10 (Krizhevsky, 2009) and ImageNet (Russakovsky et al., 2015) datasets.

**Learned scattering projections** Beyond scattering trees, the projections $P_j$ of a scattering transform (7) can be redefined as arbitrary orthogonal $1 \times 1$ convolutional operators, which reduce the number of scattering channels: $P_j P_j^T = \mathrm{Id}$. Orthogonal projectors acting along the direction index $\ell$ of wavelet filters can improve classification (Oyallon & Mallat, 2015). We are now going to learn these linear operators together with the final linear classifier. Before computing this projection, the mean and variances of each scattering channel is standardized with a batch normalization $B_N$, by setting affine coefficients $\gamma = 1$ and $\beta = 0$. This projected scattering operator can be written:

$$S_P = \prod_{j=1}^{J} P_j \, B_N \, \rho_r \, F_w.$$

Applying a linear classifier to $S_P(x)$ defines a deep convolutional network whose parameters are the $1 \times 1$ convolutional $P_j$ and the classifier weights $W, b$. The wavelet convolution filters in $F_w$ are not learned. The orthogonality of $P_j$ is imposed through the gradient steps (6) applied to $F = P_j^T$. Table 2 shows that learning the projectors $P_j$ more than halves the scattering classification error of $S_P$ relatively to $S_T$ on CIFAR-10 and ImageNet, reaching AlexNet accuracy on ImageNet, while achieving a higher Fisher ratio.

The learned orthogonal projections $P_j$ create invariants to families of linear transformations along scattering channels that depend upon scales, directions and phases. They correspond to image transformations which have been linearized by the scattering transform. Small diffeomorphisms which deform the image are examples of operators which are linearized by a scattering transform (Mallat, 2012). The learned projector eliminates within-class variabilities which are not discriminative across classes. Since it is linear, it does not improve linear separability or the Fisher ratio. It takes advantage of the non-linear separation produced by the previous scattering layers.

The operator $P_j$ is a projection on a family of orthogonal directions which define new scattering channels, and is followed by a wavelet convolution $F_w$ along spatial variables. It defines separable convolutional filters $F_w P_j$ along space and channels. Learning $P_j$ amounts to choosing orthogonal directions so that $\rho_r F_w P_j$ optimizes the class means separation. If the class distributions are invariant by rotations, the separation can be achieved with wavelet convolutions along the direction index $\ell$ (Oyallon & Mallat, 2015), but better results are obtained by learning these filters. This separable scattering architecture is different from separable approximations of deep network filters in discrete cosine bases (Ulicny et al., 2019) or in Fourier-Bessel bases (Qiu et al., 2018). A wavelet scattering computes $\rho_r F_w P_j$ as opposed to a separable decomposition $\rho_r P_j F_w$, so the ReLU is applied in a

Table 3: Evolution of Fisher ratio across layers for the scattering concentration network $S_C$ with a ReLU with threshold $\rho_{rt}$, on the CIFAR dataset.

| **CIFAR** | Layer | 0 | 1 | 2 | 3 | 4 | 5 | 6 | 7 | 8 |
|---|---|---|---|---|---|---|---|---|---|---|
| | Fisher | 1.8 | 11 | 13 | 11 | 15 | 15 | 22 | 25 | 40 |

higher dimensional space indexed by wavelet variables produced by $F_w$. It provides explicit coordinates to analyze the mathematical properties, but it also increase the number of learned parameters as shown in Table 4, Appendix D.

**Concentration along scattering channels**  A projected scattering transform can separate class means, but does not concentrate class variabilities. To further reduce classification errors, following Section 2.1, a concentration is computed with a tight frame soft-thresholding $F_j^T \rho_t F_j$, applied on scattering channels. It increases the dimension of scattering channels with a $1 \times 1$ convolutional tight frame $F_j$, applies a soft-thresholding $\rho_t$, and reduces the number of channels with the $1 \times 1$ convolutional operator $F_j^T$. The resulting concentrated scattering operator is

$$S_C = \prod_{j=1}^{J}(F_j^T \, \rho_t \, F_j) \, (P_j \, B_N \, \rho_r F_w). \tag{8}$$

It has $2J$ layers, with odd layers computed by separating means with a ReLu $\rho_r$ and even layers computed by concentrating class variabilities with a soft-thresholding $\rho_t$. According to Section 2.1 the soft-threshold is $\lambda = 1.5\sqrt{d/p}$. This soft-thresholding may be replaced by a thresholded rectifier $\rho_{rt}(u) = \max(u - \lambda, 0)$ with a lower threshold $\lambda = \sqrt{d/p}$. A logistic classifier is applied to $S_C(x)$. The resulting deep network does not include any learned bias parameter, except in the final linear classification layer. Learning is reduced to the $1 \times 1$ convolutional operators $P_j$ and $F_j$ along scattering channels, and the linear classification parameters.

Table 2 gives the classification errors of this concentrated scattering on CIFAR for $J = 4$ (8 layers) and ImageNet for $J = 6$ (12 layers). The layer dimensions are specified in Appendix D. The number of parameters of the scattering networks are given in Table 4, Appendix D. This concentration step reduces the error of $S_C$ by about $40\%$ relatively to a projected scattering $S_P$. A ReLU thresholding $\rho_{rt}$ produces an error slightly below a soft-thresholding $\rho_t$ both on CIFAR-10 and ImageNet, and this error is also below the errors of ResNet-20 for CIFAR and ResNet-18 for ImageNet. These errors are also nearly half the classification errors previously obtained by cascading a scattering tree $S_T$ with several $1 \times 1$ convolutional layers and large MLP classifiers (Zarka et al., 2020; Oyallon et al., 2017). It shows that the separation and concentration learning must be done at each scale rather than at the largest scale output. Table 3 shows the progressive improvement of the Fisher ratio measured at each layer of $S_C$ on CIFAR-10. The transition from an odd layer $2j - 1$ to an even layer $2j$ results from $F_j^T \rho_t F_j$, which always improve the Fisher ratio by concentrating class variabilities. The transition from $2j$ to $2j + 1$ is done by $P_{j+1}\rho_r F_w$, which may decrease the Fisher ratio because of the projection $P_{j+1}$, but globally brings an important improvement.

## 4 CONCLUSION

We proved that separation and concentration of probability measures can be achieved with rectifiers and thresholdings applied to appropriate tight frames $F$. We also showed that the separation of class means can be achieved by cascading wavelet frames that are not learned. It defines a scattering transform. By concentrating variabilities with a thresholding along scattering channels, we reach ResNet-18 classification accuracy on CIFAR-10 and ImageNet.

A major mathematical issue is to understand the mathematical properties of the learned projectors and tight frames along scattering channels. This is necessary to understand the types of classification problems that are well approximated with such architectures, and to prove lower bounds on the evolution of Fisher ratios across layers.

ACKNOWLEDGMENTS

This work was supported by grants from Région Ile-de-France and the PRAIRIE 3IA Institute of the French ANR-19-P3IA-0001 program. We would like to thank the Scientific Computing Core at the Flatiron Institute for the use of their computing resources.

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
