# OpenReview forum: "Separation and Concentration in Deep Networks"
_ICLR.cc/2021/Conference — ICLR 2021 Poster_

### Official Review · AnonReviewer4 · 2020-10-27
**Intriguing insights into class mean/variance evolution in deep networks**

**Rating:** 8
**Confidence:** 4

**Review:**

This is an interesting paper which seeks to explain the recently observed phenomenon in deep net image classification, whereby variance of activations for a fixed class becomes small / collapses in deeper layers, while the class means remain well-separated. The authors make an intriguing connection with the classical results of Donoho & Johnstone (1994) on soft thresholding to show that for a certain mixture of Gaussian mixtures, soft thresholding reduces the order of intra-class variation from O(d) to O(log d) while only moving the means by O(log d), provided that the means of the mixture components are compressible (approximately sparse) in the given frame.


A. Strong points

- The simplified framework proposed by the authors seems to indeed distill the key aspects of the problem and the results on soft thresholding are very promising. It is reassuring to see a connection to sparsity / compressibility which plays important but still elusive role in deep network classification.
- Theorem 2.2. which uses the soft thresholding non-linearity gives an intriguing insight into how the above reduction of variance could occur.
- The numerical experiments convincingly illustrate the theoretical discussions.
- As a meta point, I very much agree with the statement that excessive focus on universal approximation properties could blur the truly important aspects of deep networks for image classification.


B. Weak points:

- There is a bit of a discontinuity in the paper when the scattering transform is introduced. There seem to be two almost independent messages: in the first part of the paper different contractive activations are evaluted, intuition is given, theoretical results are derived for soft thresholding. In the second part of the paper a complex scattering transform is introduced. Now it is not anymore clear how to define soft thresholding or a ReLU---only the modulus is obvious. The experiments are not anymore evaluating improvements in the Fisher ratio across layers but rather demonstrating that a particular network with fixed filters and learned channel combinations classifies well. It is not clear how the findings of the first part of the paper inform the second part (save for the fact that the used wavelet frame is tight and so are the 1x1 convolutions). I would appreciate if the narrative would make a better, more organic connection between the two parts.


C. Recommendation:

I recommend that the paper be accepted. It provides an original and convincing insight into the internal mechanics of deep networks for image classification.


D. Questions / suggestions:

- In the case of soft thresholding, the bias term is implicit in the nonlinearity. It is also learned from data by computing the standard deviation of <x, f_m>. Thus it does not seem completely true that this network does not have any learnable bias terms.
- From the last paragraph of 2.1 it seems that a similar implicit bias is also used for ReLU activations. Is this true? How is the threshold \lambda set in this case? If it is true, then it seems different from certain existing bias-free networks whose linearizations are indeed linear (as opposed to affine) operators, e.g. the cited Mohan et al. (2019).
- In my opinion the first paragraph in 2.1. could use some rewriting / refactoring. The streamlining might have went to far. E.g. one wants to identify only one out of C components, x is a mixture random vector, components are equiprobable, not x. It could be friendly to the reader to write down the mixture model (that is to say the density), ... Similar remarks hold throughout the manuscript.
- Perhaps I am misreading (1), but to me it seems that one way it can be made large is that all classes have the same distribution with a large \mu_c and small covariance. I suppose the key is that one optimizes over contractive maps such as projections, which don't allow the means to grow. Is this right? Is it a motivation to use Parseval frames?
- Beyond theory, especially in the numerics, it seems to me that one could do strictly better by relaxing the tightness constraint (provided sufficient training data). Would this further improve the numerics?
- At the end of Section 2 the authors state that characterizing the mean transformations of ReLU and modulus brings about considerable difficulties. Could we have a sense of those difficulties?
- In the third paragraph under "Choice of contraction", perhaps it is worth mentioning that not all filters f_m can be bandpass as otherwise F could not be a frame.
- In 3.1, I am confused by the sentence after the last display on page 6 and the significance of "o": "Each R_j is ... by cascading more than o modulus". Could you explain this better? (nb: plural is moduli).


E. Minor comments

- full stop missing after display in Theorem 2.1
- bottom of p4: concentrations -> contractions?
- The assumption that E(\| x \|^2) = d under "Choice of contraction" should probably appear close to the first paragraph of 2.1

---

> ### Author Response · Authors · 2020-11-19
> **Answer to Reviewer 4**
>
> Thank you for your very detailed review.The weak points raised in section B of your review were addressed with several modifications in the paper:
> - You are right that there was a gap between Section 2 and Section 3 on scattering transform. This gap was filled by restructuring Section 2 and treating separately the properties of a ReLU and the soft-thresholding. Proposition 2.1, which is elementary, proves that a ReLU can only increase the Fisher ratio in a sign-invariant tight frame because it defines a non-linear operator which is linearly invertible. Section 3 now defines the scattering transform with a ReLU over a wavelet tight frame, which is also sign-invariant. This is done by representing the complex channels with a phase index, which had been done previously for texture synthesis, which we also reference.  Section 2 thus proves that this scattering transform increases the Fisher ratio, which is a key result. Interestingly, this modification also improved classification results, which now reach ResNet-18.
> - Following your recommendation, we verify numerically the increase of Fisher ratios with a scattering transform, in a new Table 3.
>
> Concerning your questions:
> - As now explained in the text, the threshold  $\lambda$ is set to be $1.5 \sqrt{d/p}$ for a soft-thresholding $\rho_t$ and $\sqrt{d/p}$ for a ReLU with threshold $\rho_{rt}$. Indeed, the standard deviation of $\langle x, f_m \rangle$ is approximately $|| f_m || \sigma$ (assuming $\mathrm{Cov}(x) = \sigma^2 \mathrm{Id}$), which is equal to $\sqrt{d/p}$ with our chosen normalizations ($\sigma=1$). The threshold $\lambda$ is thus not learned from the data but set a priori.
> - Depending upon the desired property, we may use a non-zero threshold for a ReLU, so that it does concentrate class variabilities through a thresholding, or we may set it to zero, so that the ReLU implements a pure separation operator, as in the scattering transform. In the latter case, it is similar to the work of Mohan et al. (2019).  When using a threshold $\lambda$ in the ReLU, we now explain that it is set in the same manner as for the soft-thresholding.  Setting the ReLU threshold to zero hardly harms performance (by about 2% on ImageNet).
> - We clarified the mixture notation in the text.
> - The Fisher ratio is indeed large when all classes have a large mean and small covariance. It is invariant by any invertible affine operator,  which increases in the same proportions the norm of class means and class covariances. Parseval tight frames are used because it is a normalization which simplifies the analysis of soft-thresholding concentration.
> - We observed that relaxing the tight frame constraint did not bring  a significant increase of classification accuracies.
> - A ReLU with a non-zero threshold is difficult to analyze because its effect is a mix of mean separation and concentration of class variabilities. However, as previously mentioned, a ReLU with a zero threshold is easily analyzed as shown by Proposition 2.1 in the new version. An absolute value is also difficult because it depends on whether probabilities distributions are symmetric or not, which we now mention in Section 2. If the probability distributions are symmetric, then it behaves as a ReLU with a zero threshold.
> - The frame $f_m$ must indeed contain at least one low-pass filter. We eliminated this paragraph which is not needed anymore.
> - The scattering now uses a ReLU non-linearity which is only  applied to the band-pass filters $g_{\ell,\alpha}$ as in standard scattering transform. It is not applied on the output of $g_0$ so that these linear averaging operators are cascaded which leads to wavelets of different scales. The dimension reduction operators $P_j$ (previously $R_j$) prune the branches of the scattering tree to eliminate all channels that have been transformed by more than $o$ band-pass filters and ReLU non-linearities. This is a standard procedure in scattering transforms, where $o$ is often taken to be $2$. The parameter $o$ is called the scattering order. We now explain it in more detail.
>
> We corrected the text according to your minor comments.

---

> > ### Comment · AnonReviewer4 · 2020-11-23
> > **Clarity and narrative improved**
> >
> > Thank you for addressing all of my concerns. The revised manuscript indeed reads much more cohesively. The main points are presented in a clearer way. I thus increased my score from 7 to 8. Also, very cute that my remarks had a spurious side effect of boosting the accuracy of the scattering classifier!

---

### Official Review · AnonReviewer1 · 2020-10-28
**Interesting analysis for a specific type of network**

**Rating:** 7
**Confidence:** 3

**Review:**

This work introduces the concatenation of a tight frame with a scalar non-expansive operator (mainly the modulus and soft-thresholding) as a unit, which - when applied to scattering transform coefficients - can yield high quality image classification results, improve the Fisher discriminant ratio, and (for soft-thresholding) allow some mathematical analysis in the sense of variance reduction bounds.

The paper is well-written, well-motivated, interesting and, to the best of my knowledge, novel. It combines theoretical insights with practical results that are only slightly worse than ResNet-18 on ImageNet.

On the negative side I only have a couple of minor things:
- It is slightly confusing to me that the operator $\rho$ is called "contracting" instead of "non-expansive". In the optimization literature a contraction would have to satisfy the stated property with an additional factor \gamma < 1 on the right hand side. I think it is worth rephrasing this everywhere (even in the title) to avoid confusion.
- I think it is worth stating how well the Parseval regularization enforces the orthogonality constraint for all numerical experiments. Why does $\alpha$ vary quite strongly from experiment to experiment?
- Is the batch normalization for stabilizing the learning of W' really necessary despite the rather well-behaved remaining architecture?
- On the practical side, it would have been interesting to see if the remaining gap to ResNet can be closed if the fixed spatial wavelet filters are (partially) replaced by learnable ones. To highlight the advantages of avoiding to learn such filters, the overall number of learnable parameters or the training times could be compared.

In summary, I think this is an interesting paper that does merit a publication.


----------------------
After the rebuttal: I'd like to thank the authors for their answers, particularly for resolving the confusion about the term "contraction". I believe this is a good paper and stick to my rating of recommending its acceptance.

---

> ### Author Response · Authors · 2020-11-19
> **Answer to Reviewer 1**
>
> Thank you for your thorough review, which led to important modifications of the paper.
>
> - You are right that “contraction” was confusing. As previously mentioned, we eliminated it everywhere and specified the properties of each non-linearity (ReLU, soft-thresholding, absolute values), and also changed the title.
> - We give the frame bound ratios obtained with the Parseval regularization, which are between 0.99 and 1.01 and hence nearly tight. The  Parseval regularization parameter $\alpha$ can actually be set to 0.0005 for every experiment.
> - Yes in our setting, we did observe that the final batch normalization improves the optimization and reduces the classification error, which is not well understood.
> - We have now reached ResNet-18 accuracy on ImageNet by replacing the modulus by a ReLU on wavelet coefficients at each phase. The advantage of preserving phase had been observed before in several publications on texture synthesis but not for classification.  Learning spatial filters would destroy the knowledge that we have on each scattering channel that is indexed by the wavelet scales, angles and phase, which is important mathematically.
> - We added the number of parameters of each architecture in Appendix D. The wavelet transform gives a clear mathematical structure at each layer but it does not reduce the number of parameters that are learned, because it also increases dimension by introducing angle and phase parameters.

---

### Official Review · AnonReviewer3 · 2020-10-28
**The paper suggests using tight frames with neural networks but end up with training scattering transforms**

**Rating:** 6
**Confidence:** 4

**Review:**

The paper proposes using networks that are composed of tight frames to analyze the clustering property of networks across layers.
Yet, the main focus of the paper in the first part is to construct the tight frame-based networks and then in the second part to train scattering transforms based networks.
While the ideas are interesting I have several concerns:

1. The idea of encouraging tight frame structure is not new and appeared already in several works in the literature.
2. The idea of training a scattering transform is not new and has been done before. For example, there is a work by Mallat that shows that one may just add 1x1 convs to the scattering transform and train it. So the current work is not of much difference.

Given these two concerns, I don't think the current novelty is sufficient for publication.

=========================================================================================

Updated review:
Thank you for the clarification. The previous version was indeed confusing to me. I have raised my score although I think some points still need to be addressed in the revision following my previous comments as they were not fully addressed nor in the response neither in the revision:
1.  The concern with respect to previous works is not only regarding Parseval networks. There are other more recent works that use orthogonality constraints on the network. Such examples include
https://ieeexplore.ieee.org/document/8877742
https://openaccess.thecvf.com/content_cvpr_2017/papers/Xie_All_You_Need_CVPR_2017_paper.pdf
https://proceedings.neurips.cc/paper/2018/hash/bf424cb7b0dea050a42b9739eb261a3a-Abstract.html

All these works show a similar observation to the one claimed in the paper that by using orthogonality (or frame-like) operators one may train a network without skip connections and get similar results.
Indeed, in the paper, more observations are being made that are different than what is presented in these works but a more proper comparison should be made.

2. This is the work the authors should look at by Mallat
https://arxiv.org/pdf/1809.06367.pdf
They get similar performance to ResNet with a scattering transform-based network.
Indeed, also here it is not exactly the same network that the authors here are using but there are remarkable similarities and these should be well addressed.

---

> ### Author Response · Authors · 2020-11-19
> **Answer to Reviewer 3**
>
> Thank you for your review. We disagree on your assessment of the novelty of this work but your remarks helped us to modify the paper to clarify this aspect.
>
> You are right that tight frame structures have been studied in Parseval networks. We referenced it in the original paper and we now emphasize it better in the introduction. However, what is new is the *mathematical use* of this property which allows us to prove a concentration bound of within-class variability with a soft-thresholding on Gaussian mixture models, and guarantee the increase of Fisher ratios with rectifiers.
>
> You are also right that several works have incorporated learned layers with the scattering transform, but *after* the scattering transform. In this work, it is the projectors *within each layer* of the scattering transform which are learned. It leads to much higher classification rates. The previous work that you refer to could only reach AlexNet accuracy with a two or three hidden layer MLP. In this work we reach ResNet-18 accuracy, which reduces the Top 5 error by 10%, with a linear classifier! Moreover, this new scattering structure has a simpler mathematical interpretation, where the learned projectors implement linearized invariants along scattering channels.
>
> We thank the reviewer for his remarks and have added the mentioned references to the paper, to better illustrate the key differences. We therefore believe the reviewer is mistaken in its assessment of the novelty of this work.

---

### Official Review · AnonReviewer2 · 2020-10-30
**Interesting Paper**

**Rating:** 6
**Confidence:** 3

**Review:**

The paper introduces structured deep network architectures
that can be analyzed mathematically and have high classification accuracies on
complex image databases. The proposed mechanism consists of iterating over tight frame contractions.
They also show that spatial filters do not need to be learned, and can be defined from wavelet frames.

############

Overall inclined for accepting the paper although I am a bit hesitant due to the lack of experimental details and/or code. Overall, the
 main idea is interesting and novel and the paper is well written.

############

Pros

* Interesting idea and theoretical results/implications.

############

Cons
* Details on the experiments are largely missing, i.e. parameters etc are not listed anywhere in the experimental section.
* No code provided or even mentioned in the paper. This makes the experimental verification harder.

---

> ### Author Response · Authors · 2020-11-19
> **Answer to Reviewer 2**
>
> Thank you for your review.
>
> The parameters needed to reproduce the experiments (sizes of each layer, learning rate schedule, scattering parameters, Parseval regularization) are provided in the Appendix D in the supplementary material. We have now included the url where the code will soon be released to reproduce all experiments.

---

### Author Response · Authors · 2020-11-19
**General announcement**

We thank the reviewers for the time and effort they spent on their reviews. This is more than a polite thanks because these remarks helped us to greatly improve the paper. We made important modifications to address several remarks:

- Following the recommendation of Reviewer 1 we suppressed everywhere (and in the title) the word “contraction” that was confusing. In the title we replaced it by separation and concentration which are the main properties that are studied.
- To address the comment of Reviewer 3 we clarified and modified the introduction and the contributions.
- A major point is that we now fully relate the mathematics of the scattering transform in Section 3 to results in Section 2 as recommended by Reviewer 4. This required to modify Section 2, and make different paragraphs to analyze class mean separation with rectifiers and concentration with soft-thresholdings. Section 3 was adjusted so that the scattering transform is  now a particular case of class-mean separation with a ReLU applied to wavelet tight frames. Wavelet complex coefficients are now represented with reals indexed by a phase, to which we apply a ReLU.
- With a scattering computed with a ReLU we now reach the accuracy of ResNet-18 on CIFAR and ImageNet, which is excellent news!

Please find detailed answers to your reviews below.

---

### Decision · Program_Chairs · 2021-01-07
**Final Decision**

**Decision:**

Accept (Poster)

**Comment:**

After reading the author’s response, all reviewers recommend accepting the paper.

The authors provided an extensive response carefully considering all reviewers' comments. After incorporating the feedback, the manuscript improved in terms of presentation, relation to the literature and empirical results.

The paper is very well written and motivated. On top of the insightful analysis, experimental results are strong, obtaining comparable performance to that of a ResNet-18 on ImageNet.

R1 and R3 strongly support the paper while R2 and R4 consider it borderline.

R2 raised questions about experimental details and reproducibility. While R2 did not comment, these concerns were very clearly addressed by the authors in the view of the AC.

R4 was initially concerned with the novelty of the approach, but changed their mind after the author's response. The AC encourages the authors to further consider the feedback provided by the reviewer after the discussion period was over.